# Fatigue and resilience in Master's and PhD students in the Covid-19 pandemic in Brazil: A cross-sectional study

Izabel Alves das Chagas Valóta[1]*, Rafael Rodrigo da Silva Pimentel[2], Ana Paula Neroni Stina Saura[1], Rodrigo Marques da Silva[3], Ana Lucia Siqueira Costa Calache[1‡], Marcelo José dos Santos[2‡]

1 Department of Medical-Surgical Nursing, Nursing School of University of São Paulo, São Paulo, São Paulo, Brazil, 2 Professional Guidance Department, Nursing School of University of São Paulo, São Paulo, São Paulo, Brazil, 3 Sena Aires Faculty of Science and Education, Valparaíso de Goiás, Goiás, Brazil

☯ These authors contributed equally to this work.
‡ ALSCC and MJS also contributed equally to this work.
* izabel.chagas@usp.br

## Abstract

The aim of this study was to analyze levels of fatigue and resilience of Brazilian graduate students during the COVID-19 pandemic and to determine whether there is an association between fatigue and resilience and sociodemographic and academic factors. Data were analyzed using descriptive and inferential statistics, and it was discovered that the variables associated with higher levels of resilience were age; having children; being retired; receiving income above five minimum wages; having had greater problems in other phases of the research schedule; coming from private universities; being from the north of Brazil; studying the area of Health; and having their research schedule unaffected during the pandemic. On the other hand, lack of resilience was associated with not having children; being less well-off financially; being younger; being a woman; studying in a public university; and having to postpone part of the research during the pandemic. The conclusion of the study indicated the need for graduate programs to design strategies to deal with fatigue and promote resilience in Master's and PhD students.

## 1. Introduction

The pandemic caused by the SARS-Cov-2 infection (COVID-19) in 2019 brought great challenges that resulted in the need for adaptation in all spheres of society and whose consequences could be observed in the functioning of institutions and social practices throughout the world. In higher education institutions this reality was no different, and institutions attempted to preserve the safety of their community and maintain public health care. Protocols were therefore adopted that transferred the traditional model of teaching and learning to a completely online form. Traditional work practices were also replaced by the remote model, and public spaces for social interaction were closed [1].

Given this scenario, graduate students, who were already in a challenging environment, marked by psychological pressure with instability [2], now had to deal with considerable

**Data Availability Statement:** All relevant data are within the paper and its Supporting information files.

**Funding:** Valóta IAC - 88887.356470/2019-00 - "This study was financed in part by the Coordenação de Aperfeiçoamento de Pessoal de Nível Superior – Brasil (CAPES) – Finance Code 001". https://www.gov.br/capes/pt-br Pimentel RRS - 88887.508751/2020-00 - "This study was financed in part by the Coordenação de Aperfeiçoamento de Pessoal de Nível Superior – Brasil (CAPES) – Finance Code 001". https://www.gov.br/capes/pt-br The funders had no role in study design, data collection and analysis, decision to publish, or preparation of the manuscript.

**Competing interests:** The authors have declared that no competing interests exist.

changes in guidance and financial uncertainty. With the closure of physical spaces at universities, it was necessary to cancel long-term experiments and interrupt or delay the data collection period [1], change the class schedule, qualification exams, and dissertation and thesis presentations, which were now carried out remotely. In addition, graduate students had to deal with the family, personal, and emotional consequences of the new limits imposed by Covid restrictions [3], which may have impacted students' mental health and led to fatigue.

The demands present in this situation have been found to lead to the development of mental disorders, especially in individuals who are psychologically more vulnerable [2, 4]. The mental health problems of graduate students have received attention in recent years, with the development of systematic reviews and primary studies reporting the negative psychological effects on the health of these individuals [5, 6]. A study carried out in Belgium indicates that one in two PhD students experiences psychological distress and one in three is at risk of a common psychiatric illness [7]. In the US, a study showed that 90% of doctoral students are six times more likely to have anxiety and depression compared to the general population [8].

Fatigue can be seen as a universal phenomenon, experienced by healthy and sick people. There is a consensus that it is a subjective, multifactorial and multidimensional phenomenon, understood as an unpleasant physical sensation, with cognitive and emotional components, manifested in a feeling of tiredness that is not relieved by the usual energy restoration strategies [9–11]. Manifestations of fatigue involve a decrease in self-care, physical capacity, memory and concentration, lack of interest and motivation in activities, weakness, irritability, frustration, sadness and spiritual anguish [12, 13].

Fatigue is a crucial factor to be identified in the training of students as it can affect academic success and learning [14, 15]. In the literature, fatigue has been investigated under different conditions. Among university students it has been identified as severe [15], and among doctoral students in Chiropractic [14], Medical, Social and Natural Sciences, Languages and Literatures, Mathematics and Computer Sciences it was at a moderate level [16].

In recent decades, positive psychology has emerged as an alternative in the search for psychological resources to deal with the onset of fatigue, anxiety and depression [17, 18]. Psychological constructs such as self-efficacy, resilience, hope, and optimism have played positive roles in alleviating fatigue symptoms [19].

A study carried out with doctors in China showed a high incidence of fatigue among the professionals interviewed, with resilience being negatively associated with fatigue [20]. Similar findings emerged in another study of hospital workers in South Korea, which showed that lower levels of resilience were positively associated with symptoms of physical fatigue [21].

Resilience can be conceptualized as a unidimensional or multidimensional construct [22]. Given the various definitions, several concepts have emerged [23, 24], including resilience as a personal quality that moderates the impact of negative stressors; as a necessary quality to deal with adverse and traumatic situations [22, 25]; and as an element in the interaction between the individual and biological, cognitive, interpersonal and contextual factors [22–24]. Evidence indicates that resilience can also affect perseverance, student empowerment [26] and academic success [27–30].

Fatigue and resilience are thus two phenomena that deserve attention among students, especially in challenging periods such as those of the COVID-19 pandemic. These two phenomena are present in the daily lives of students and have been little studied among graduate students in Brazil. Therefore, the following questions arose: what were the fatigue and resilience levels of master's and doctoral students during the Covid-19 pandemic in Brazil? Was there an association between sociodemographic and academic factors with the performance of fatigue and resilience levels? This study therefore aims to analyze the level of fatigue and

resilience among Brazilian Master's and PhD students during the COVID-19 pandemic and to determine whether there is an association with sociodemographic and academic factors.

## 2. Method

This is an observational and cross-sectional study.

### 2.1 Population and recruitment

Only those graduate students aged over 18 years, enrolled in Brazilian educational institutions and who responded to all survey items, were included in this study. Students who did not answer all questions in the survey were excluded.

Data collection was carried out in online format from July to August 2020. The dissemination and invitation to participate in the research were carried out on Facebook® and Linkedin® through posts in research and graduate groups. Emails identified in a government repository were sent to 4,648 Master's and PhD graduate programs throughout Brazil, with a request to circulate the survey link among students.

### 2.2 Measures

**2.2.1 Questionnaire with sociodemographic and academic data.** The collected sociodemographic and academic data questionnaire addressed age, sex, marital status, children, region of Brazil and state, monthly income, supplementary income, current course, area of knowledge, institution, Ministry of Education graduate program grade, source of income, whether monthly income was sufficient to support oneself, impact of COVID-19 on family income and on the progress of the research schedule, and phase of greatest impact. See a complete version of the questionnaire in S1 Questionnaires.

**2.2.2 The Revised Piper Fatigue Scale (PFS).** The revised Piper Fatigue Scale (PFS), developed by Barbara Piper [9] and validated for use in Brazil [13, 31], assessed fatigue in a multidimensional way. It had an internal consistency of 0.94, and for the subscales it ranged between 0.84 and 0.94 (Cronbach's alpha) [31]. In this study internal consistency presented values in the Behavioral dimensions of 0.884; Affective 0.905; Sensory 0.913; and Cognitive 0.897.

The original revised scale contained 22 items distributed in four dimensions: Behavioral/ Intensity; Affective; Sensory; and Cognitive/Humour. Each dimension receives a score that corresponded to the average of the scores for each item and varies from zero to ten. Clinically, in order to consider the presence of fatigue, a score greater than four was adopted, considering the total score [31]. In addition to the 22 scored items, there were five open questions (Items 1, and 24 to 27) that were not used to calculate the instrument score. In this study, in addition to the 22 items, Item 1 was used to verify the fatigue time.

**2.2.3 The Visual Analog Fatigue Scale (VAFS).** To assess the level of fatigue, the VAFS of pain intensity adapted for fatigue was used [32]. The scale went from 0 to 10 cm, with higher scores representing greater severity or intensity of fatigue, while lower scores represented mild levels of fatigue. The fatigue cut-off point was considered as mild = 1–2; moderate = 3–6 and intense = 7–10 [32].

**2.2.4 The Wagnild and Young Resilience Scale.** The Wagnild and Young Resilience Scale, validated in Brazil [33], was used to measure resilience levels of positive psychosocial adaptation when confronting important events. It contained 25 items on a Likert-type scale ranging from one (strongly disagree) to seven (strongly agree). It presented two factors in the original version: Factor 1 "Personal Competence"; and Factor II "Acceptance of Self and Life" [25, 33]. It had an internal consistency of 0.80 (Chronbach's alpha) [25]. In this study in

"Personal Competence" it was 0.885 and in "Acceptance of Self and Life" 0.693. The resilience scale score was obtained by the sum of the total responses of the 25 items ranging from 25 to 175 points, and the higher the score, the higher the individual's resilience [25, 33]. A result below 120 was considered "low resilience"; between 121 and 145 "moderately low to moderate resilience"; and above 145 "moderate-high to high resilience" [34].

### 2.3 Data analysis

Data were collected and organized in the Research Electronic Data Capture (REDcap) software [35] and analyzed in the R software statistical package (Version 3.6.1) and in the Statistical Package for the Social Sciences (SPSS) (Version 20.0). Qualitative variables were presented as absolute (N) and relative (%) frequencies, and quantitative variables were presented as mean, standard deviation, and confidence interval (95% CI). The internal consistency of the PFS and the resilience scale were analyzed, and Cronbach's alpha coefficients were determined considering values between 0.65 and 0.70 as acceptable, 0.70 and 0.80 as good, and 0.90 as very good.

In the bivariate analysis, non-parametric Mann-Whitney, Brunner-Munzel and Kruskal-Wallis statistical tests were applied to analyze the association between sociodemographic and numerical academic variables and the level of fatigue and resilience. To assess the effect size of the tests, the following classification was adopted: null (0 to 0.10); weak (0.11 to 0.29); moderate (0.30 to 0.49); strong ($\geq$ 0.50) [36].

Pearson's correlation coefficient was used to calculate the relationshi11p between fatigue and resilience in the applied scales and their dimensions and to correlate age with levels of fatigue and resilience. The following correlation coefficients (r) were established: 0.00–0.19 very weak correlation; 0.20–0.39 weak correlation; 0.40–0.59 moderate correlation; 0.60–0.79 strong correlation; and 0.80–1.00 very strong correlation [37].

The application of the Bonferroni correction on the values of the hypothesis tests adjusted the significance level for this study to 0.36% (0.0036).

### 2.4 Ethical aspects

This study was approved by the Research Ethics Committee of the School of Nursing at the University of São Paulo, Brazil (Opinion number: 4,420,446/2020), meeting national and international research standards.

## 3. Results

Of the 5,492 forms accessed, 3,331 (60.6%) fully completed the survey. The age of the participants ranged from 21 to 70 years, with a mean of 32.4 years (SD: 7.68). Most students were female (70.28%); with a partner (62.08%); without children (78.93%); residing in the Southeast region of Brazil (49.86%). 21.66% of the total number of female students had children. Income ranged from one to two times the minimum wage (47.76%), with the majority receiving some type of scholarship (49.29%), but most did not receive additional income during the pandemic (83.40%). 50.53% reported insufficient income to support themselves. However, 59.44% considered that there was no impact on their income during the pandemic (Table 1).

The majority of the students were from Master's courses (50.65%) and public universities (92.22%); and a large number were from the area of health sciences (23.18%). There was a predominance of graduate programs, evaluated by the government agency responsible for evaluation of graduate programs with grades 4 [good] and 5 [very good] (59.20%). Most students (84.39%) reported an impact on the research schedule during the pandemic period, with the data collection and analysis phases (53.53%) suffering the greatest impact (Table 2).

**Table 1. Distribution of Master's and PhD students, according to sociodemographic data (N = 3,331) Brazil, 2023.**

| Variables | N | % |
|---|---:|---:|
| **Sex** | | |
| Female | 2,341 | 70.28 |
| Male | 990 | 29.72 |
| **Marital status** | | |
| With partner | 2,068 | 62.08 |
| Without partner | 1,263 | 37.92 |
| **Children** | | |
| Yes | 702 | 21.07 |
| No | 2,629 | 78.93 |
| **Region of Brazil** | | |
| Centre-west | 283 | 8.50 |
| Northeast | 544 | 16.33 |
| North | 76 | 2.28 |
| Southeast | 1,661 | 49.86 |
| South | 767 | 23.03 |
| **Income[a]** | | |
| 1 to 2 minimum wages | 1,591 | 47.76 |
| 3 to 4 minimum wages | 1,002 | 30.08 |
| > 5 minimum wages | 738 | 22.16 |
| **Source of income** | | |
| Retired | 10 | 0.30 |
| Scholarship | 1,642 | 49.29 |
| Family resources | 253 | 7.60 |
| Fixed employment | 1,208 | 36.27 |
| Informal employment | 209 | 6.27 |
| Others* | 9 | 0.27 |
| **Complementary income[b]** | | |
| Yes | 553 | 16.60 |
| No | 2,778 | 83.40 |
| **Income sufficient to live on** | | |
| Yes | 1,648 | 49.47 |
| No | 1,683 | 50.53 |
| **Impact of COVID-19 on income** | | |
| Yes | 1,351 | 40.56 |
| No | 1,980 | 59.44 |

*Others: 2—Unemployment benefit; 3—Self-employed; 4—Help from the government during the pandemic.

[a]Family income based on the value of a minimum wage = R$1,045.00 / US$193.96 –rate on February 2, 2021.

[b] Complementary income provided by the Federal Government during the pandemic R$600.00/US$111.36.

The students had an average PFS score of 6.08 (SD = 2.04, CI = 6.01–6.15), which indicates the presence of fatigue. In the Behavioral dimension, the mean score was 6.53 (SD = 2.23, CI = 6.46–6.61), Affective 5.87 (SD = 2.46, CI = 5.79–5.95), Sensory 5.96 (SD = 2.43, CI = 5.88–6.05), Cognitive 5.89 (SD = 2.21, CI = 5.82–5.97). The mean fatigue time reported by 62.53% of the students was 8.5 months; 14.56% reported 5.3 weeks; 7.41% 46.7 days; for 3.27% it averaged 0.2 hours; and 1.53% of students reported 22.3 minutes.

**Table 2. Distribution of Master's and PhD students, according to course characteristics and research impact (N = 3,331) Brazil, 2023.**

| Variables | N | % |
|---|---|---|
| **Course** | | |
| Master's | 1,687 | 50.65 |
| PhD | 1,644 | 49.35 |
| **Area of knowledge[a]** | | |
| Health Sciences | 772 | 23.18 |
| Human Sciences | 496 | 14.89 |
| Applied Social Sciences | 410 | 12.31 |
| Exact and Earth Sciences | 331 | 9.94 |
| Biological Sciences | 324 | 9.73 |
| Engineering | 303 | 9.10 |
| Agricultural Sciences | 263 | 7.90 |
| Linguistics, Languages and Literature, and Arts | 233 | 6.99 |
| Multidisciplinary | 199 | 5.97 |
| **University** | | |
| Private | 259 | 7.78 |
| Public | 3,072 | 92.22 |
| **Grade of graduate course [b]** | | |
| 1 to 3 [poor] | 470 | 14.11 |
| 4 to 5 [good and very good] | 1,972 | 59.20 |
| 6 to 7 [excellent] | 889 | 26.69 |
| **Impact of COVID-19 on the research schedule** | | |
| Yes | 2,811 | 84.39 |
| No | 520 | 15.61 |
| **Phase of greatest impact** | | |
| Data analysis and collection | 1,503 | 53.53 |
| Courses taken | 443 | 15.78 |
| Project and qualification exam | 583 | 20.76 |
| Final writing | 260 | 9.26 |
| Others* | 19 | 0.68 |

*Others: 7—Internship during the international doctorate (sandwich PhD); 3—Presentation of the dissertation or thesis; 5—Adaptation to the course; 2—Publication of articles; 2—Consultation of bibliography.

[a]Nomenclatures of academic areas of knowledge according to the Brazilian National Council for Scientific and Technological Development (CNPq). According to the classification, "Health Sciences" includes Medicine, Nutrition, Dentistry, Pharmacy, Nursing, Public Health, Physical Education, Speech Therapy, Physical Therapy, and Occupational Therapy.

[b]Graduate programs evaluated by the Coordination for the Improvement of Higher Education Personnel (CAPES), a specific government agency that defines the concept of the program, ranging from 1 [low] to 7 [high], taking into account the scientific production of students and advisors, among other aspects.

The students showed moderate fatigue on the VAFS (6.10; SD = 2.51, CI = 6.02–6.19). Variation of results is observed between the two fatigue scales, and the average fatigue value is described according to Brazilian states (Fig 1).

Students had a mean resilience score of 126.00 (SD = 19.81; CI = 125.56–126.90), which indicates levels of "moderately low to moderate resilience", with the mean resilience value shown according to Brazilian states in Fig 2.

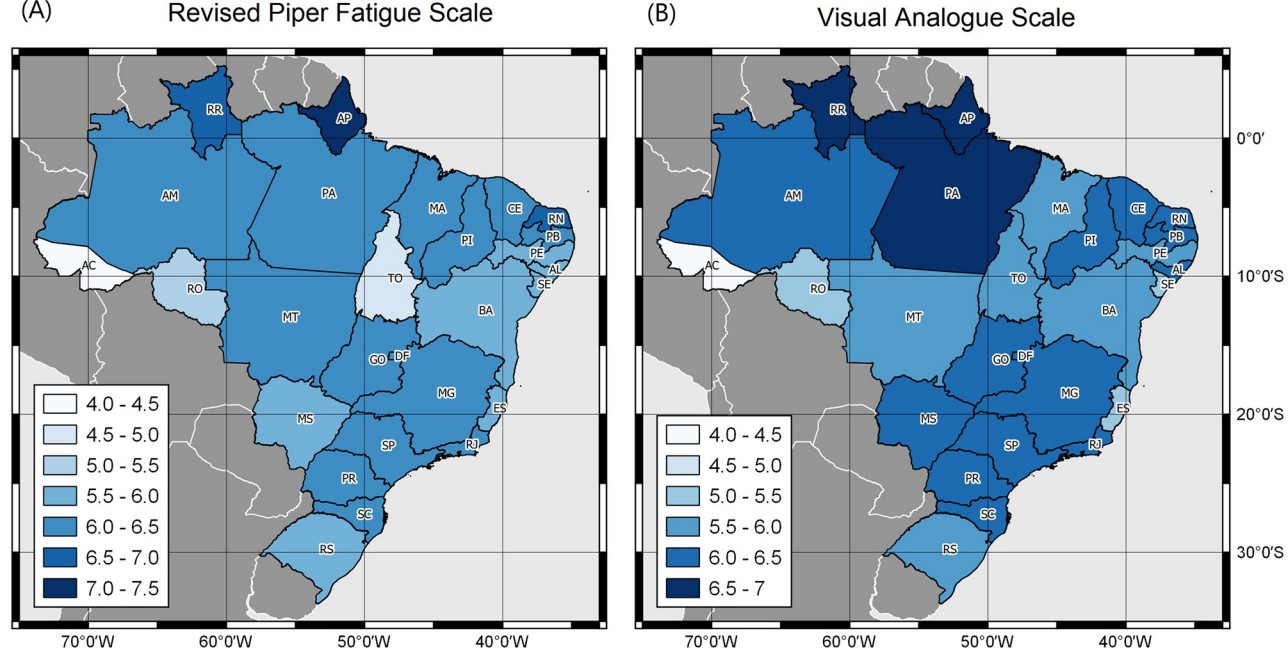

**Fig 1. Distribution of the average Fatigue score among graduate students, according to Brazilian states, Brazil, 2023.** Presentation of the Revised Piper Fatigue Scale (FFS) (A) and the Visual Analogue Scale (VAS) (B). Darker colors represent higher levels of fatigue in individuals. Map created in free and open source software QGIS® Version 3.24.1. For the limits of states and countries, the base of the Brazilian Institute of Geography and Statistics (IBGE/DGC). Continuous Cartographic Base of Brazil, 1:250.000 –BC250: version 2017. Rio de Janeiro, 2017 - https://www.ibge.gov.br/geociencias/cartas-e-maps/bases-cartograficas-continuas/15759-brasil.html?=&t=access-to-product).

In the Competence dimension, the mean resilience score was 90.64, ranging from 24 to 119 (SD = 14.52, CI = 90.14–91.13); in Acceptance, the mean score was 35.36, ranging from 12 to 56 (SD = 7.31, CI = 35.03–35.53).

The inferential analysis showed that women had higher mean levels of fatigue (p<0.001) on both scales. Age showed a weak positive correlation with the mean level of resilience (r = 0.221, p<0.001), very weak negative correlation in the PFS (r = -0.106, p<0.001) and in the VAFS (r = -0.077, p<0.001). Students from the northern region of Brazil showed higher levels of resilience (p<0.001).

Students with children also showed higher levels of resilience (p<0.001). Having an income which is at least five times greater than the minimum wage is correlated with greater resilience (p<0.001), whereas students with income between one and two minimum wages showed greater fatigue (p<0.001) in the evaluation on both scales. Having retirement as a source of income favored resilience (p<0.001); and financial dependence on family resources made students more fatigued (p = 0.002) on the PFS. Impact on income due to the pandemic favored higher levels of fatigue (p<0.001), shown on both scales (Table 3).

Students from private universities, who suffered no impact on their research schedule, were more resilient (p<0.001). Students enrolled in public universities had higher levels of fatigue in the PFS (p = 0.001); and students whose research schedule was affected had higher levels of fatigue (p<0.001) on both scales.

In the area of Health Sciences students whose research schedule was affected in other phases were more resilient (p<0.001); however, in the area of Agricultural Sciences, those who were in the phase of the final writing of the dissertation or thesis were more fatigued (p<0.001).

# Resilience

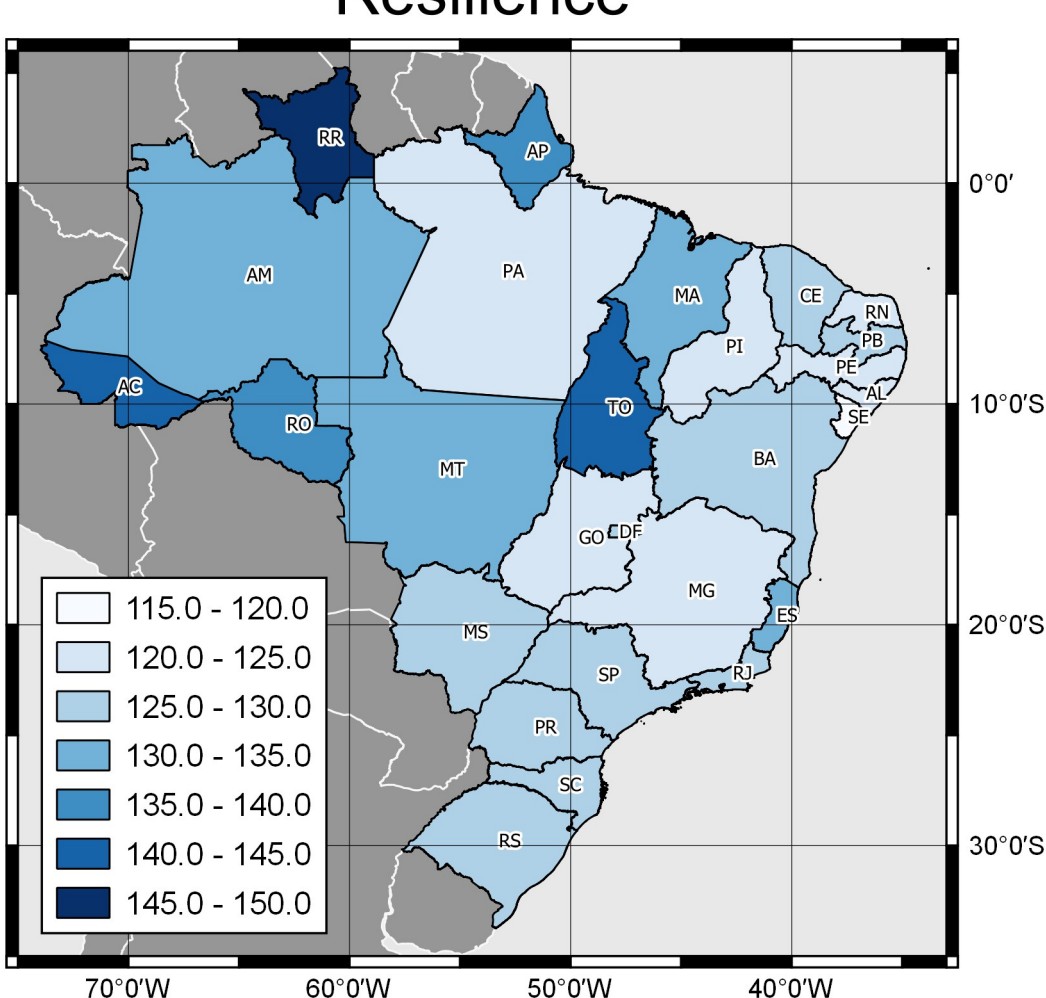

**Fig 2. Distribution of the average resilience score among graduate students, according to Brazilian states, Brazil, 2023.** Darker colors represent higher levels of resilience in individuals. Map created in free and open source software QGIS® Version 3.24.1. For the limits of states and countries, the base of the Brazilian Institute of Geography and Statistics (IBGE/ DGC). Continuous Cartographic Base of Brazil, 1:250.000 –BC250: version 2017. Rio de Janeiro, 2017 - https://www.ibge. gov.br/geociencias/cartas-e-maps/bases-cartograficas-continuas/15759-brasil.html?=&t=access-to-product).

There was no significant statistical impact on the following variables: supplementary income in the pandemic, marital status, course, and grade of the graduate program (Table 4).

The variables with moderate negative correlations with resilience were the sensory (r = - 0.427) and cognitive (r = - 0.431) dimensions. The variables with weak negative correlations with resilience were the behavioral (r = - 0.370) and affective (r = - 0.338) dimensions.

The two resilience dimensions were negatively correlated with the four fatigue dimensions. The variables with the weakest negative correlation with the competence dimension included the general level of fatigue (r = - 0.395), behavioral (r = - 0.319), affective (r = - 0.296), sensory (r = - 0.388), and cognitive (r = - 0.385) dimensions. In the acceptance dimension there was a moderate negative correlation with the level of fatigue (r = - 0.424), with the cognitive dimension (r = - 0.405) and a weak negative correlation with the behavioral (r = - 0.371), affective (r = - 0.328), and sensory (r = - 0.384) dimensions.

**Table 3. Mean fatigue and resilience scores by sociodemographic factors, Brazil, 2023.**

| Variables | Fatigue–PFS | | | Fatigue–VAFS | | | Resilience | | |
|---|---|---|---|---|---|---|---|---|---|
| | M | SD | p-value (ES) | M | SD | p-value (ES) | M | SD | p-value (ES) |
| **Sex** | | | | | | | | | |
| Female | 6.24 | 1.99 | < 0.001[a] (0.12) | 6.31 | 2.44 | < 0.001[b] (0.12) | 125.37 | 19.99 | 0.008[a] |
| Male | 5.68 | 2.12 | | 5.62 | 2.60 | | 127.47 | 19.32 | |
| **Marital status** | | | | | | | | | |
| With partner | 6.03 | 2.06 | 0.105[a] | 6.07 | 2.52 | 0.376[a] | 126.70 | 19.56 | 0.007[a] |
| Without partner | 6.15 | 2.02 | | 6.16 | 2.49 | | 124.85 | 20.18 | |
| **Children** | | | | | | | | | |
| Yes | 5.86 | 2.21 | 0.010[b] | 5.98 | 2.73 | 0.520b | 132.37 | 18.36 | < 0.001b (0.18) |
| No | 6.13 | 1.99 | | 6.14 | 2.44 | | 124.29 | 19.84 | |
| **Region of Brazil** | | | | | | | | | |
| Centre-west | 6.38 | 1.95 | 0.095[c] | 6.32 | 2.42 | 0.580[c] | 125.64 | 20.09 | < 0.001[c] (0.19) |
| Northeast | 6.06 | 2.13 | | 6.01 | 2.56 | | 125.20 | 20.03 | |
| North | 5.94 | 2.22 | | 6.11 | 2.54 | | 134.92 | 20.01 | |
| Southeast | 6.09 | 1.97 | | 6.14 | 2.45 | | 125.54 | 19.22 | |
| South | 5.97 | 2.13 | | 6.02 | 2.62 | | 126.79 | 20.59 | |
| **Income** | | | | | | | | | |
| 1 to 2 minimum wages | 6.31 | 1.96 | < 0.001[c] (0.13) | 6.32 | 2.38 | < 0.001[c] (0.08) | 122.90 | 20.48 | < 0.001[c] (0.20) |
| 3 to 4 minimum wages | 5.96 | 2.07 | | 5.98 | 2.56 | | 126.89 | 19.52 | |
| > 5 minimum wages | 5.72 | 2.12 | | 5.80 | 2.66 | | 131.46 | 17.30 | |
| **Source of income** | | | | | | | | | |
| Retired | 3.95 | 1.40 | 0.002[c] (0.22) | 4.00 | 2.11 | 0.075[c] | 147.40 | 13.53 | < 0.001[c] (0.25) |
| Scholarship | 6.13 | 1.95 | | 6.10 | 2.40 | | 124.10 | 19.63 | |
| Family resources | 6.28 | 2.00 | | 6.17 | 2.46 | | 121.53 | 21.44 | |
| Fixed employment | 5.98 | 2.10 | | 6.09 | 2.61 | | 129.46 | 18.79 | |
| Informal employment | 6.11 | 2.35 | | 6.24 | 2.70 | | 125.35 | 21.41 | |
| Others* | 5.84 | 2.54 | | 5.56 | 3.05 | | 124.22 | 25.83 | |
| **Complimentary income** | | | | | | | | | |
| Yes | 6.11 | 1.90 | 0.959[b] | 6.09 | 2.46 | 0.698[a] | 124.58 | 20.23 | 0.072[a] |
| No | 6.07 | 2.07 | | 6.11 | 2.51 | | 126.28 | 19.72 | |
| **Impact of COVID-19 on income** | | | | | | | | | |
| Yes | 6.27 | 2.05 | < 0.001[a] (0.03) | 6.31 | 2.50 | < 0.001[a] (0.01) | 125.98 | 20.40 | 0.826[b] |
| No | 5.94 | 2.03 | | 5.97 | 2.50 | | 126.01 | 19.40 | |

PFS = Revised Piper Fatigue Scale. VAFS = Visual Analog Fatigue Scale. ES = Effect Size. Tests:

[a]Mann-Whitney;

[b]Brunner-Munzel;

[c]kruskal-Wallis.

*Others: Unemployment insurance; self-employed; complementary income provided by the Federal Government during the pandemic R$600.00/US$111.36.

Scores on the Resilience Scale and PFS assessments showed a moderate negative correlation, suggesting that higher levels of resilience were associated with lower levels of fatigue (r = - 0.446, CI = -0.473 to—0.418, p<0.001).

## 4. Discussion

This study assessed the impact of the COVID-19 pandemic on graduate students in Brazil and some key instruments to analyze fatigue and resilience. These two constructs were highlighted

**Table 4. Mean fatigue and resilience scores by academic factors, Brazil, 2023.**

| Variables | Fatigue–PFS | | | Fatigue–VAFS | | | Resilience | | |
|---|---|---|---|---|---|---|---|---|---|
| | M | SD | p-value (ES) | M | SD | p-value (ES) | M | SD | p-value (ES) |
| **Course** | | | | | | | | | |
| Master's | 6.13 | 2.04 | 0.097[a] | 6.13 | 2.49 | 0.597[a] | 125.33 | 20.28 | 0.113[b] |
| PhD | 6.02 | 2.04 | | 6.08 | 2.52 | | 126.68 | 19.29 | |
| **University** | | | | | | | | | |
| Public | 6.11 | 2.03 | 0.001[a] (0.17) | 6.13 | 2.49 | 0.036b | 125.50 | 19.81 | < 0.001[a] (0.10) |
| Private | 5.69 | 2.14 | | 5.76 | 2.72 | | 131.91 | 18.88 | |
| **Area of knowledge** | | | | | | | | | |
| Health Sciences | 5.79 | 2.17 | | 5.84 | 2.65 | | 129.71 | 18.80 | |
| Human Sciences | 6.04 | 2.04 | | 5.96 | 2.55 | | 124.30 | 20.40 | |
| Applied Social Sciences | 6.08 | 1.97 | | 6.27 | 2.47 | | 126.81 | 19.17 | |
| Exact and Earth Sciences | 6.10 | 1.93 | | 6.09 | 2.36 | | 124.68 | 18.77 | |
| Biological Sciences | 6.34 | 1.96 | < 0.001[c] (0.04) | 6.28 | 2.36 | 0.010[c] | 123.24 | 21.19 | < 0.001[c] (0.03) |
| Engineering | 5.98 | 1.98 | | 6.02 | 2.46 | | 126.20 | 18.89 | |
| Agricultural Sciences | 6.36 | 2.05 | | 6.37 | 2.41 | | 125.24 | 20.66 | |
| Linguistics, Languages and Literature, and Arts | 6.34 | 1.88 | | 6.30 | 2.46 | | 121.44 | 20.11 | |
| Multidisciplinary | 6.25 | 2.14 | | 6.46 | 2.49 | | 126.87 | 20.44 | |
| **Grade of graduate course** | | | | | | | | | |
| 1 to 3 [poor] | 5.97 | 2.14 | 0.057[c] | 5.95 | 2.60 | 0.006[c] | 127.28 | 19.76 | 0.085[c] |
| 4 to 5 [good and very good] | 6.13 | 2.05 | | 6.21 | 2.50 | | 126.04 | 20.26 | |
| 6 to 7 [excellent] | 6.01 | 1.97 | | 5.96 | 2.47 | | 125.21 | 18.79 | |
| **Impact of COVID-19 on research schedule** | | | | | | | | | |
| Yes | 6.23 | 1.97 | < 0.001[b] (0.09) | 6.29 | 2.41 | < 0.001[b] (0.07) | 125.13 | 19.76 | < 0.001[a] (0.00) |
| No | 5.23 | 2.24 | | 5.11 | 2.77 | | 130.69 | 19.41 | |
| **Phase of greatest impact** | | | | | | | | | |
| Collection and analysis of data | 6.18 | 1.96 | < 0.001[c] (0.21) | 6.20 | 2.41 | < 0.001[c] (0.25) | 125.50 | 19.80 | < 0.001[c] (0.18) |
| Courses taken | 5.94 | 2.09 | | 5.95 | 2.55 | | 127.45 | 18.98 | |
| Elaboration of project and qualification exam | 6.47 | 1.87 | | 6.60 | 2.27 | | 123.33 | 20.44 | |
| Final writing of dissertation or thesis | 6.62 | 1.79 | | 6.89 | 2.20 | | 122.44 | 18.70 | |
| Others* | 4.77 | 2.39 | | 4.21 | 2.62 | | 132.53 | 21.32 | |

PFS = Revised Piper Fatigue Scale. VAFS = Visual Analog Fatigue Scale. ES = Effect Size. Tests:

[a]Mann-Whitney;

[b]Brunner-Munzel;

[c]kruskal-Wallis.

*Others: International mobility; Thesis presentation; Adaptation to the course; Publication of articles; Consultation of the bibliography.

as they can affect student performance and learning [14, 15, 30]. Our results indicate that academic and socioeconomic aspects were associated with higher levels of fatigue and resilience.

The pandemic had an impact on the research schedule and the completion of students' theses and dissertations. Concern for one's own health and that of loved ones, difficulty concentrating, changes in sleep patterns and a reduction in social interactions [38, 39] were stressors that may have led students with difficulties in the final writing stage of their dissertation or thesis to present greater fatigue.

Studies carried out in the pre-pandemic period indicated the relationship between fatigue and academic activities [14, 16] and showed that graduate students were already fatigued, which may have increased in the pandemic. On the other hand, similar to our results, a study

with graduate students from all over Brazil during the pandemic showed that 72.0% of students made changes to their projects [40], which may have jeopardized the other phases of their research schedules, similar to those evidenced in this study. This may have caused both psychosocial and academic frustrations, leading to adaptive behaviors and resilience [41]. Adjustments in expectations between the training that students envisioned and what was possible during the pandemic may have been a positive strategy for their performance in terms of resilience.

The areas of knowledge present different ways of working with in terms of research and knowledge development. In health sciences, students were more likely to adapt to changes, face challenges and see themselves as people capable of overcoming difficulties [42, 43], and this was related to resilience. Knowledge of forms of transmission of other diseases and prevention mechanisms may have contributed to these individuals having greater emotional balance to deal with and adapt positively to adversity. On the other hand, students in the area of agricultural sciences had higher levels of fatigue. The reasons that may support such a finding have already been shown as the pandemic meant that students had to interrupt and/or modify their research [40], which in this area is mostly experimental, often including cultivation and field and laboratory planting [44].

In public universities in Brazil there is a constant demand for academics or researchers to increase their scientific production as they are under pressure from government classification processes which rank institutions and graduate programs on their research productivity [45]. Researchers or academics must then prioritize their work in teaching, research and extension activities. If they are to be promoted and be competitive for jobs then they must produce publications and have success in obtaining research funding. This pressure and bureaucratic structure can cause an overload on academics which is transferred to and has a direct impact on graduate students.

This is in contrast to private universities in Brazil where there is less demand for research publications and a greater focus on teaching. Academics are often on more balanced workload allocations with contracts which are not solely based on research. Graduate students are also in better financial situations and this is correlated with resilience. But it should be stated that research in private universities is very limited, and less than eight percent of students in this survey were from private universities.

The economic aspect also impacts the lives of graduate students as many depend on grants for their subsistence and/or depend on the financial support of their families. These resources may be insufficient for maintaining themselves. However, obtaining other sources of income is difficult because graduate studies require a large amount of time for students to carry out their academic activities. The pandemic worsened this situation as the monetary loss of students or their families and the search for another job during the pandemic were elements that led to higher levels of fatigue [46, 47]. Graduate students who had economic difficulties were identified as more fatigued.

The studied population is predominantly female and does not have children; however, the variable 'having children' was associated with resilience. Children can be a source of motivation to face adverse situations [48]. This fact can also be linked to the performance of academic resilience, defined as "the dynamic process and interaction between an academic and their constantly changing environment that uses available internal and external resources to produce positive results in response to different contextual challenges, environmental and development issues" [49, p.13]. In this definition, emphasis is given to the relational characteristic of resilience when considering social interactions to the detriment of the focus on the individual [50].

The study found that women are more fatigued than men. Other studies also reached the same result [14, 51–54]. Care must be taken with the interpretation of this result in order to avoid a superficial narrative in discussions of sex and gender as the interaction of ecological, family, behavioral and physiological mechanisms can drive decisions related to gender health, exposure to risks and biological vulnerabilities [55]. In this sense, men can minimize fatigue due to their masculine traits of stoicism and resistance [56], while women express and communicate feelings, emotions and symptoms more easily [57, 58]. The environment should also be considered as social characteristics such as the division of labor by gender and exposure to institutionalized sexism can be drivers of fatigue [52]. During the pandemic, graduate students had to deal with an increase in tasks and work hours, which may have contributed to greater emotional exhaustion.

Age was an important determining factor for both fatigue and resilience. Older students, who had experienced more adversities throughout their lives, had greater resilience [59–61]. Younger graduate students were more fatigued, a fact that differs from a study carried out before the pandemic, in which the age of students was not a significant factor for fatigue [14]. During the pandemic, young people may have had fewer of their own and/or institutional resources to deal with psychic stressors. The lack of a support network and social interactions for many young people may have been a risk factor for fatigue.

The results show that there is a need to implement university programs that favor the improvement of resilience among students so that they can overcome the challenges present during their time as graduate students. This would result in the most demanding situations being experienced with less suffering and without the characteristic repercussions of fatigue. Some strategies for improving resilience are psychotherapeutic approaches; resilience training [24]; mindfulness-based workshops [62]; and group intervention [63]. Other strategies for coping with psychological distress in the face of COVID-19 carried out by universities were physical exercise; Zoom meetings with friends from religious communities; staying connected with family, friends and professors; and establishing self-help techniques such as meditation, relaxation, listening to music, and staying positive) [64, 65].

Fatigue and resilience can vary in traumatic or more intense situations; however, this general assessment of students' resilience and fatigue provides a good basis to work from. It is also noteworthy that resilience was a mediator for fatigue as increasing one reduces the other and vice versa. In addition, this study demonstrates the association between resilience and fatigue in graduate students during the COVID-19 pandemic in Brazil and is therefore a breakthrough for Brazilian and world science in understanding how these two phenomena develop in critical times for society.

A limitation of this study was the non-homogeneous sample as there are differences in the proportions of graduate students in various regions of Brazil.

## 5. Conclusion

Most students had significant and moderate average levels of fatigue and moderately low levels of resilience. The higher the level of resilience, the lower the levels of fatigue. Sociodemographic and academic factors were associated with these phenomena. Understanding the relationships between graduate students' individual characteristics and external factors can be important for developing strategies to deal with fatigue and improve students' resilience. The COVID-19 pandemic may have intensified fatigue and required individuals, in the face of adversity, to develop resilience.

## Supporting information

**S1 Database.**
(XLSX)

**S1 Questionnaires. Questionnaire on sociodemographic and academic data.**
(DOCX)

## Acknowledgments

The authors would like to thank all graduate students who voluntarily participated in the study and all graduate programs that helped to publicize this research.

## Author Contributions

**Conceptualization:** Izabel Alves das Chagas Valóta, Rafael Rodrigo da Silva Pimentel, Ana Paula Neroni Stina Saura.

**Data curation:** Izabel Alves das Chagas Valóta, Rafael Rodrigo da Silva Pimentel, Ana Paula Neroni Stina Saura.

**Formal analysis:** Izabel Alves das Chagas Valóta, Rafael Rodrigo da Silva Pimentel.

**Funding acquisition:** Izabel Alves das Chagas Valóta, Rafael Rodrigo da Silva Pimentel.

**Investigation:** Izabel Alves das Chagas Valóta, Rafael Rodrigo da Silva Pimentel.

**Methodology:** Izabel Alves das Chagas Valóta, Rafael Rodrigo da Silva Pimentel, Ana Paula Neroni Stina Saura, Rodrigo Marques da Silva, Ana Lucia Siqueira Costa Calache, Marcelo José dos Santos.

**Project administration:** Ana Lucia Siqueira Costa Calache, Marcelo José dos Santos.

**Resources:** Izabel Alves das Chagas Valóta, Rafael Rodrigo da Silva Pimentel.

**Software:** Izabel Alves das Chagas Valóta, Rafael Rodrigo da Silva Pimentel, Ana Paula Neroni Stina Saura.

**Supervision:** Izabel Alves das Chagas Valóta, Rafael Rodrigo da Silva Pimentel, Ana Lucia Siqueira Costa Calache, Marcelo José dos Santos.

**Validation:** Izabel Alves das Chagas Valóta, Rafael Rodrigo da Silva Pimentel, Ana Paula Neroni Stina Saura, Rodrigo Marques da Silva, Ana Lucia Siqueira Costa Calache, Marcelo José dos Santos.

**Visualization:** Izabel Alves das Chagas Valóta, Rafael Rodrigo da Silva Pimentel, Ana Paula Neroni Stina Saura, Rodrigo Marques da Silva.

**Writing – original draft:** Izabel Alves das Chagas Valóta, Rafael Rodrigo da Silva Pimentel, Ana Paula Neroni Stina Saura, Rodrigo Marques da Silva, Ana Lucia Siqueira Costa Calache, Marcelo José dos Santos.

**Writing – review & editing:** Izabel Alves das Chagas Valóta, Rafael Rodrigo da Silva Pimentel, Ana Paula Neroni Stina Saura, Rodrigo Marques da Silva, Ana Lucia Siqueira Costa Calache, Marcelo José dos Santos.

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
