## [Decision Letter · Decision Letter 0]

4 Feb 2022

PONE-D-21-28536Fatigue and resilience in master's and doctoral students during the COVID-19 pandemic in Brazil: a cross-sectional studyPLOS ONE

Dear Dr. Valóta,

Thank you for submitting your manuscript to PLOS ONE. After careful consideration, we feel that it has merit but does not fully meet PLOS ONE’s publication criteria as it currently stands. Therefore, we invite you to submit a revised version of the manuscript that addresses the points raised during the review process. I have carefully considered the comments of both reviewers and read your manuscript.  One reviewer has recommended rejection – but the grounds for this are not clear.  It appears some cultural differences may have led them to make decisions based on color and gender statements.  Please read this with consideration – they were meant in good faith – but have not completely been sensitively articulated.  Overall the result section of the manuscript needs significant improvement.  You have used some interesting tools to determine fatigue and resilience.  The result section needs more coherency and less fragmentation.  One way to reduce this is to decrease the number of sub headings.  Also more literature is needed in the discussion – include literature on gender and other areas which support your findings.       Please submit your revised manuscript by Mar 21 2022 11:59PM. If you will need more time than this to complete your revisions, please reply to this message or contact the journal office at plosone@plos.org. Please include the following items when submitting your revised manuscript:A rebuttal letter that responds to each point raised by the academic editor and reviewer(s). You should upload this letter as a separate file labeled 'Response to Reviewers'.A marked-up copy of your manuscript that highlights changes made to the original version. You should upload this as a separate file labeled 'Revised Manuscript with Track Changes'.An unmarked version of your revised paper without tracked changes. You should upload this as a separate file labeled 'Manuscript'.

We look forward to receiving your revised manuscript.

Kind regards,

Pauline M. Ross, PhD

Academic Editor

PLOS ONE

Journal Requirements:

3. We note that Figures 1 and 2 in your submission contain [map/satellite] images which may be copyrighted. All PLOS content is published under the Creative Commons Attribution License (CC BY 4.0), which means that the manuscript, images, and Supporting Information files will be freely available online, and any third party is permitted to access, download, copy, distribute, and use these materials in any way, even commercially, with proper attribution. For these reasons, we cannot publish previously copyrighted maps or satellite images created using proprietary data, such as Google software (Google Maps, Street View, and Earth). For more information, see our copyright guidelines: http://journals.plos.org/plosone/s/licenses-and-copyright.

a. You may seek permission from the original copyright holder of Figures 1 and 2 to publish the content specifically under the CC BY 4.0 license.  

Reviewers' comments:

Reviewer's Responses to Questions

**Comments to the Author**

1. Is the manuscript technically sound, and do the data support the conclusions?

Reviewer #1: No

Reviewer #2: Partly

2. Has the statistical analysis been performed appropriately and rigorously? 

Reviewer #1: No

Reviewer #2: I Don't Know

3. Have the authors made all data underlying the findings in their manuscript fully available?

Reviewer #1: No

Reviewer #2: Yes

4. Is the manuscript presented in an intelligible fashion and written in standard English?

Reviewer #1: Yes

Reviewer #2: Yes

5. Review Comments to the Author

Reviewer #1: Please see my uploaded comments. However, as I am forced to also use this box for some reason, I have copied the comments below:

The authors of this study raise and attempt to address an intriguing question – how did the pandemic affect postgraduate students? To achieve this, an impressively large scale questionnaire was disseminated to students throughout Brazil, resulting in over 3000 responses. Unfortunately, many of the conclusions drawn appear to be very shallow in nature and I worry about the lack of novelty presented in the article (e.g. it comes as little surprise that individuals with more financial support and whose research remained unimpacted were better off). Additionally, there appear to some statistical flaws and deeply concerning demographic questions being utilised, which overall detract from the study. I will detail such examples below, but I am sadly unconvinced that this article should be published. Lastly, your introduction is not sufficient with only one page provided to cover the entire area of study relating to the difficulties of post-graduate work (which is a large field in and of itself!).

To begin, I wish to raise the issue of your demographic questions. Without seeing the questionnaire itself (which is absent from the appendix), I can’t fully confirm this, but it appears that you asked participants if their race/colour was Yellow, White, Indigenous, Brown or Black. Not only is this deeply inappropriate, but you did not even raise that you were investigating race in your methodology. As you ultimately find no statistical relevance here, I found strongly recommend removing this data. As for why this is inappropriate, ‘Yellow’ is a deeply offensive term often used to combine many distinct Asian identities (e.g. Korean, Japanese, Chinese, Taiwanese etc.). Furthermore, did you only ask if the participants were ‘male’ or ‘female’? What of non-binary or trans people? As it stands, I can’t actually confirm what you asked but I suspect you may have fallen back onto old stereotypes.

Following on from these stereotypes, I worry about many of the conclusions drawn. You state ‘Being a woman often involves the accumulation of work and domestic functions’. While it is true that societal conventions often result in women undertaking more housework, I fail to see how this is related to their biological sex of gender identity. As written, it implies that the two are deeply connected, rather than an outcome of patriarchal values. Furthermore, you provide no references for this comment.

The next statement of concern was ‘Not having children can be an element that, by reducing the social interaction and emotional support brought by the mother-child relationship, can increase the chance of fatigue in women’. I am unsure why you feel that the lack of a parent-child support structure was only relevant to women? As written, are you suggesting that such a connection is less relevant to fathers? I see no reason why such a specific gender-based correlation should exist. This is compounded by an error is your statistical analysis, in which the p cut-off value does not appear to be corrected for the number of items utilised in your questionnaire. A Bonferroni correction requires you to divide the p value by the number of items in your questionnaire, which likely lowered the cut-off to <0.01 (the significance you saw for this comparison) and thereby may invalidate this analysis. Additionally, you provide no literature based justification for the use of correlation values as effect sizes. Traditionally, each statistical test has an associated effect size calculation which it appears you have chosen to not use.

Next on the list of statements that need justification is ‘Therefore, a study suggests that government officials periodically review the effectiveness of the isolation measures implemented and consider ways to make such measures more flexible in order not to compromise the health of the population [16] or consider the need for emergency aid as an aid to maintain the minimum family income’. I am surprised by the implication here that government agencies are not already taking this into account. As such, and this relates to my overall concern around the lack of novelty in this article, I’m not sure what this statement adds to the topic area of interest.

You have also stated that ‘Another important aspect is that biological science students may have been more resilient, as they faced challenges inherent to the benchtop researcher [19] and maintained good results in the resilience score’. I’m sorry, but I’m not sure what you mean by this. Are not all bench researchers facing challenges ‘inherent to the benchtop researcher’? How is the experience of Biological Science students unique? Surely not all of them experienced the frontline of the pandemic as you raise for the Health Science students? In fact, what are these grouped together and, indeed, why and how were any of the groups combined (i.e. why are the Exact Sciences and Earth, Engineering, and Agricultural Sciences combined into one?).

I have two last comments before I wrap up. Firstly, the lack of acknowledgements is surprising to me, as I would have at the very least expected an extension of gratitude to the students who took the time to complete the questionnaire. Secondly, the writing of this article, while generally reasonable, occasionally contained grammatical errors or signs of a lack of detailed proofreading. Some examples are below:

1) Therefore, a study suggests that government officials periodically review the effectiveness of the isolation measures … (I presume you mean - therefore, this study suggests …)

2) … such measures more flexible in order not to compromise the health of the population … (I presume you mean mental health).

Overall, without a full justification of the statistical testing undertaken, the removal of stereotypical assumptions, the presence of the actual questionnaire used AND the novelty of this work better highlighted, I unfortunately cannot recommend this work for publication.

Reviewer #2: Comments for authors:

Overall. The study analyses the relationship between fatigue and resilience while controlling for academic and sociodemographic factors for postgraduate students in Brazil. The findings and the contribution could be strengthened by linking the discussion better to your data, and discussing what already happens in universities in terms of resilience training. Many programs already offer co-curricular and extra-curricular workshops and seminars on time management, mindfulness, etc. all of which I would argue might be helpful in mitigating fatigue and learning resilient behaviours. This is not considered in the current manuscript version. Also, it is not clear to me whether you found any particular factors to be the major ‘dealbreakers’ or ‘deal-makers’. You suggest financial hardships might be a significant factor, perhaps even more important in causing stress and fatigue than lack of social support. I would strongly recommend that you highlight and emphasise your main findings in the discussion. Overall, the paper would benefit from professional editing. A few expressions are unclear in what they mean or refer to. I hope further detailed comments below will be helpful in revising the paper.

Intro:

Page13line66: expression “behaviours that promote protection’ is odd, resilience is not a behaviour per se, rather a quality or attribute I would argue, what might be examples of resilient behaviours?

Method:

Consider explaining in a few words what a prospective study is.

Why is sense strict in italics?

P4l82-83: “All of the students who did not fully respond to the data collection” perhaps simplify to who did not answer all survey questions…

L89 Survey link capitalised

Discussion:

Overall, I suggest revising the structure to better orient the reader through your argument. Currently this section is somewhat disjointed. For example, the last paragraph on ‘resilience building strategies seems disconnected and not well integrated with the rest.

P15l295; change to impact OF increased levels of stress and anxiety

P16l312-313: which Brazilian study do you refer to here: “Different results from those found in a Brazilian study showed that 56% of students were doctoral students”

P17l338-340: the role of this sentence is unclear: “It is expected that younger students suffer more from fatigue and have a less resilient behaviour pattern when compared to older ones. “ Who expects that? Does your data speak to this directly? If so, you need to make the connection to your research clearer here. And who are the researchers in line 340, you or other research?

How would you explain the finding that females who were financially disadvantaged by covid-19 experienced more fatigue but students with children were more resilient. This question comes up naturally for the reader when reading the abstract. Consider explaining this contrasting finding.

P18l371: “marked devaluation” do you mean that academic activities became less of a preference for them? Who devalued such activities?

P18l375: What do you mean by ‘intense adversity in this region’ here?

Limitations:

You state that finding other studies on this subject was difficult but you refer to several similar studies throughout.

P20l4090411, how realistic is it to conduct similar studies before pandemic, perhaps use a more nuanced language and say a general assessment of students’ resilience and fatigue would provide a good baseline or basis to work with.

Conclusion:

Review the second sentence and consider splitting in two.

6. PLOS authors have the option to publish the peer review history of their article (what does this mean?). If published, this will include your full peer review and any attached files.

Reviewer #1: No

Reviewer #2: No

---

## [Author Response · Author response to Decision Letter 0]

22 Apr 2022

Reviewer 1

The authors of this study raise and attempt to address an intriguing question – how did the pandemic affect postgraduate students? To achieve this, an impressively large-scale questionnaire was disseminated to students throughout Brazil, resulting in over 3000 responses. Unfortunately, many of the conclusions drawn appear to be very shallow in nature and I worry about the lack of novelty presented in the article (e.g. it comes as little surprise that individuals with more financial support and whose research remained unimpacted were better off). 

The conclusions made were discussed and supported internationally. It is important to emphasize that a quality study does not necessarily need to have new information. In addition, our study advances scientific knowledge by presenting the relationship between fatigue and resilience in a population that deserves greater attention.

Additionally, there appear to some statistical flaws and deeply concerning demographic questions being utilised, which overall detract from the study.

 We have made the necessary adjustments.

 I will detail such examples below, but I am sadly unconvinced that this article should be published. Lastly, your introduction is not sufficient with only one page provided to cover the entire area of study relating to the difficulties of post-graduate work (which is a large field in and of itself!).

We have made additions to the Introduction.

To begin, I wish to raise the issue of your demographic questions. Without seeing the questionnaire itself (which is absent from the appendix), I can’t fully confirm this, but it appears that you asked participants if their race/colour was Yellow, White, Indigenous, Brown or Black. Not only is this deeply inappropriate, but you did not even raise that you were investigating race in your methodology. As you ultimately find no statistical relevance here, I found strongly recommend removing this data. 

We have added the instruments used in the survey.

We have taken out the data on race and colour.

As for why this is inappropriate, ‘Yellow’ is a deeply offensive term often used to combine many distinct Asian identities (e.g. Korean, Japanese, Chinese, Taiwanese etc.). 

We emphasize that in Brazilian culture, the term “yellow” is not offensive. In Brazil, the Brazilian Institute of Geography and Statistics (IBGE), the government agency responsible for carrying out a population census every decade, conducts a survey on the colour/race of Brazilians based on self-declaration, which when asked about their colour/race according to the following options: white, black, mixed race, indigenous or yellow (https://cnae.ibge.gov.br/en/component/content/article/95-7a12/7a12-vamos-know-o-brasil/ our-people/16049-color-or-race.html)

Furthermore, did you only ask if the participants were ‘male’ or ‘female’? What of non-binary or trans people? As it stands, I can’t actually confirm what you asked but I suspect you may have fallen back onto old stereotypes.

We only considered biological sex.

Following on from these stereotypes, I worry about many of the conclusions drawn. You state ‘Being a woman often involves the accumulation of work and domestic functions’. While it is true that societal conventions often result in women undertaking more housework, I fail to see how this is related to their biological sex of gender identity. As written, it implies that the two are deeply connected, rather than an outcome of patriarchal values. Furthermore, you provide no references for this comment.

We have rewritten the paragraph and supported it with literature on gender.

The next statement of concern was ‘Not having children can be an element that, by reducing the social interaction and emotional support brought by the mother-child relationship, can increase the chance of fatigue in women’. I am unsure why you feel that the lack of a parent-child support structure was only relevant to women? As written, are you suggesting that such a connection is less relevant to fathers? I see no reason why such a specific gender-based correlation should exist. 

We have rewritten the analysis and taken out this paragraph.

This is compounded by an error in your statistical analysis, in which the p cut-off value does not appear to be corrected for the number of items utilised in your questionnaire. A Bonferroni correction requires you to divide the p value by the number of items in your questionnaire, which likely lowered the cut-off to <0.01 (the significance you saw for this comparison) and thereby may invalidate this analysis. Additionally, you provide no literature-based justification for the use of correlation values as effect sizes. Traditionally, each statistical test has an associated effect size calculation which it appears you have chosen to not use. 

 We have applied Bonferroni correction to adjust the values of hypothesis tests and have added a justification based on the literature for the use of effect size values.

Next on the list of statements that need justification is ‘Therefore, a study suggests that government officials periodically review the effectiveness of the isolation measures implemented and consider ways to make such measures more flexible in order not to compromise the health of the population [16] or consider the need for emergency aid as an aid to maintain the minimum family income’. I am surprised by the implication here that government agencies are not already taking this into account. As such, and this relates to my overall concern around the lack of novelty in this article, I’m not sure what this statement adds to the topic area of interest.

At the time of writing the manuscript, there were no standardized government measures. We have modified the paragraph.

As for the “lack of novelty” note, to date, we have not found any study of this magnitude which has evaluated more than 3,000 graduate students in Brazil during the pandemic. It is worth mentioning that when we carry out research, we do not necessarily find new data, which does not detract from the merit of the study.

You have also stated that ‘Another important aspect is that biological science students may have been more resilient, as they faced challenges inherent to the benchtop researcher [19] and maintained good results in the resilience score’. I’m sorry, but I’m not sure what you mean by this. Are not all bench researchers facing challenges ‘inherent to the benchtop researcher’? How is the experience of Biological Science students unique? Surely not all of them experienced the frontline of the pandemic as you raise for the Health Science students? 

We have rewritten the paragraph.

In fact, what are these grouped together and, indeed, why and how were any of the groups combined (i.e. why are the Exact Sciences and Earth, Engineering, and Agricultural Sciences combined into one?) 

 We have regrouped according to the classification of areas of knowledge of the National Council for Scientific and Technological Development (CNPq), an entity linked to the Ministry of Science, Technology and Innovation to encourage research in Brazil. (https://www.gov.br/capes/pt-br/centrais-de-conteudo/TabelaAreasConhecimento_072012_atualizada_2017_v2.pdf).

I have two last comments before I wrap up. Firstly, the lack of acknowledgements is surprising to me, as I would have at the very least expected an extension of gratitude to the students who took the time to complete the questionnaire.

Participants in this study were acknowledged by the authors of the work during data collection. In the manuscript, we now thank the graduate students who participated in our research.

Secondly, the writing of this article, while generally reasonable, occasionally contained grammatical errors or signs of a lack of detailed proofreading. Some examples are below:

1) Therefore, a study suggests that government officials periodically review the effectiveness of the isolation measures … (I presume you mean - therefore, this study suggests …) 

2) … such measures more flexible in order not to compromise the health of the population … (I presume you mean mental health). 

A professional editor has made the necessary changes.

 Overall, without a full justification of the statistical testing undertaken, the removal of stereotypical assumptions, the presence of the actual questionnaire used AND the novelty of this work better highlighted, I unfortunately cannot recommend this work for publication.

We have rewritten the complete justification of the statistical tests performed, removed the assumptions considered stereotyped, added the questionnaires, and highlighted the new elements of this work.

Reviewer 2

The study analyses the relationship between fatigue and resilience while controlling for academic and sociodemographic factors for postgraduate students in Brazil. The findings and the contribution could be strengthened by linking the discussion better to your data, and discussing what already happens in universities in terms of resilience training. Many programs already offer co-curricular and extra-curricular workshops and seminars on time management, mindfulness, etc. all of which I would argue might be helpful in mitigating fatigue and learning resilient behaviours. This is not considered in the current manuscript version. Also, it is not clear to me whether you found any particular factors to be the major ‘dealbreakers’ or ‘deal-makers’. You suggest financial hardships might be a significant factor, perhaps even more important in causing stress and fatigue than lack of social support. I would strongly recommend that you highlight and emphasise your main findings in the discussion. Overall, the paper would benefit from professional editing. A few expressions are unclear in what they mean or refer to. I hope further detailed comments below will be helpful in revising the paper. 

We have restructured the discussion and have been helped by a professional editor.

Intro:

Page13 line 66: expression “behaviours that promote protection’ is odd, resilience is not a behaviour per se, rather a quality or attribute I would argue, what might be examples of resilient behaviours? 

We have changed the paragraph and aligned it with the definition of resilience used by Wagnild and Young (Wagnild GM, Young HM. Development and psychometric evaluation of resilience scale. J Nurs Meas. 1993;1(2):165-78.)

Method:

Consider explaining in a few words what a prospective study is. 

We have rewritten the paragraph/sentence.

Why is sense strict in italics? 

A formatting error has been corrected.

P4l82-83: “All of the students who did not fully respond to the data collection” perhaps simplify to who did not answer all survey questions…

We have made the suggested change.

L89 Survey link capitalised.

We have rewritten the paragraph/sentence. 

Limitations:

You state that finding other studies on this subject was difficult but you refer to several similar studies throughout.

P20l4090411, how realistic is it to conduct similar studies before pandemic, perhaps use a more nuanced language and say a general assessment of students’ resilience and fatigue would provide a good baseline or basis to work with. 

 We agree with the point and have rewritten the paragraph in the limitations of the study.

Conclusion:

Review the second sentence and consider splitting it into two.

We have rewritten the paragraph/sentence as suggested.

---

## [Decision Letter · Decision Letter 1]

30 Aug 2022

PONE-D-21-28536R1Fatigue and resilience in Master's and PhD students in the Covid-19 pandemic in Brazil: a cross-sectional studyPLOS ONE

Dear Dr. Valóta,

Thank you for submitting your manuscript to PLOS ONE. After careful consideration, we feel that it has merit but does not fully meet PLOS ONE’s publication criteria as it currently stands. Therefore, we invite you to submit a revised version of the manuscript that addresses the points raised during the review process.

Could you please make the changes as required by the reviewer so that this review can be completed.  I notice the reviewer comments that there are sections which still are not logical in flow.  This is very important to fix in this version.  Following the revision, I will complete what I hope is a final read and edit – so that this manuscript can be resolved.  This very much depends however on the quality of the next revised version.  I encourage you to persist and perhaps obtain some assistance to produce a quality manuscript.  Best wishes and I look forward to the resolution. 

We look forward to receiving your revised manuscript.

Kind regards,

Pauline M. Ross, PhD

Academic Editor

PLOS ONE

Reviewers' comments:

Reviewer's Responses to Questions

**Comments to the Author**

1. If the authors have adequately addressed your comments raised in a previous round of review and you feel that this manuscript is now acceptable for publication, you may indicate that here to bypass the “Comments to the Author” section, enter your conflict of interest statement in the “Confidential to Editor” section, and submit your "Accept" recommendation.

Reviewer #2: All comments have been addressed

Reviewer #3: (No Response)

2. Is the manuscript technically sound, and do the data support the conclusions?

Reviewer #2: Partly

Reviewer #3: Yes

3. Has the statistical analysis been performed appropriately and rigorously? 

Reviewer #2: I Don't Know

Reviewer #3: Yes

4. Have the authors made all data underlying the findings in their manuscript fully available?

Reviewer #2: Yes

Reviewer #3: Yes

5. Is the manuscript presented in an intelligible fashion and written in standard English?

Reviewer #2: Yes

Reviewer #3: Yes

6. Review Comments to the Author

Reviewer #2: (No Response)

Reviewer #3: This study provides a large cross-sectional study to assess the impact of the COVID-19 pandemic on postgraduate students in Brazil. The authors should be commended in collecting and analysing a large body of data that provides a snapshot of some key instruments for analysis to assess the impact of fatigue and resilience. I can see from the response to the reviewers that the authors have made a considerable effort to review and adjust based on the feedback provided. Having not read the original submission, I can only comment directly on the revision that has been submitted.

Overall, the manuscript has had much more description and link to previous studies, which does enhance the study, however, in places, the information does lack coherence and flow. There seem to be some paragraphs that are quite short (see first paragraph of introduction) that could be linked better throughout. The 7th paragraph of the introduction, the final sentence needs re-wording – “Literature data” is not correct to refer to and you should include what the significant effects were (greater/improved?).

Your overall aim, in addition to analysing the level of fatigue and resilience, is to also consider the association with sociodemographic and academic factors, which I feel is what is lacking in the discussion. You do a good job of discussing the literature around fatigue and resilience, however, given you present two key figures associated with demographic data, I would like to see more detail around this. The novelty of your paper is that you are assessing Brazilian students – the demographics play a significant role here and this is the novelty in my opinion. This will allow you to utilise the nice studies you have to support your findings in other countries.

Whilst fatigue is not my field, I am curious as to why both a PFS and a VAFS scale are used? I may be misinterpreting this, but they seem to give slightly differing results? You also only seem to primarily discuss the PFS results. I also was unclear (for VAFS) what best result versus worst result meant? If these are common terms for these scales, please ignore, if not, please edit what this means.

In 2.3 you refer to n number as (n), however in each table it is referred to as N – please edit.

Paragraph prior to table 2 starts with “The students” – I assume you mean Most students? Or “The survey found that…”

The figure legends for the 2 images are underwritten. Whilst it is good you have included the source, you should also provide the reader with a description of what the images represent. For example, darker colour means more fatigue etc. I also recommend that figure 1 be given an A and B for each image to help the reader.

The discussion is generally good (see comments above), but I do feel it lacks flow. I am also slightly confused with paragraph 2, where you refer to females having family commitments may have significant affects on their careers, however in your study the majority of respondents were female and did not have children? Please review this data in relation to your outcomes. It would also be interesting to see what % of the female respondents did have children.

Similar to a previous reviewer, there is mention of many measures currently in place – I suggest you restructure this paragraph to be towards the latter section of the discussion in order to link this in with what you are also suggesting is implemented across the country.

For your age data, you mention this differs from the current literature – do you have a suggestion as to why your data may differ?

4.1 – I’m not sure you require the strong points section – this should be the focus of the discussion and come through to the reader without having to state it.

Overall, whilst the language has been improved, I would suggest further proof reading and structural/grammar checks prior to resubmission. Thank you for your interesting study and I wish you well with the revision.

7. PLOS authors have the option to publish the peer review history of their article (what does this mean?). If published, this will include your full peer review and any attached files.

Reviewer #2: No

Reviewer #3: No

---

## [Author Response · Author response to Decision Letter 1]

2 Mar 2023

Academic Editor

Could you please make the changes as required by the reviewer so that this review can be completed. I notice the reviewer comments that there are sections which still are not logical in flow. This is very important to fix in this version. Following the revision, I will complete what I hope is a final read and edit – so that this manuscript can be resolved. This very much depends however on the quality of the next revised version. I encourage you to persist and perhaps obtain some assistance to produce a quality manuscript. Best wishes and I look forward to the resolution.

Response: We have made substantial changes to the manuscript in response to reviewers' requests and presented an improved version.

Reviewer 2:

Thank you for revising the manuscript. I still have concerns about the suitability of this paper for publication. Reading Reviewer 1 comments and the authors’ responses further support my concerns. I will leave it to Reviewer 1 to check the technical aspects of the analysis and how the authors responded. I also echo Reviewer 1 concern about the statements made in regard to ‘yellow’. I do not think that this is appropriate for an international audience and might be highly offensive to international readers, although I take the authors’ response on board about the use of this term by Brazilian government agencies. 

Response: Thank you for the comments. We believe that the manuscript has publication potential as we could not find in the literature an extensive assessment of fatigue and resilience among graduate students in such a challenging scenario. We considered the suggestions of Reviewer 1 in the previous round, and, as you can see in the manuscript, the race/yellow color variable has been removed.

The authors are also well advised to provide a more reflective and considerate response in regard to traditional gender roles, as per Reviewer 1 suggestions.

Response: We have reconsidered traditional gender roles and our results. We have reformulated the text in the Discussion.

Please also note that my original comments on the Discussion section were not at all addressed in the authors’ response. 

Response: We apologize for not addressing your previous points in the response letter and only changing them throughout the manuscript. In this round we address the different points.

I have re-read the Discussion again and have further points for authors to consider: The writing still needs a major revision throughout the manuscript, perhaps with the help of a Native English speaker? Some examples include:

‘food products’ e.g. ‘food’?

‘bringing higher levels of fatigue’ e.g. ‘causing’?

‘Overcoming the “publish or perish” mentality’, is an odd expression

“an Indian study [69] with the same theme”, a study in India? Do you mean research questions when you say ‘theme’? 

Response: We have revised the text with the help of a native speaker.

“Graduate students who carry out their studies at private universities in Brazil may feel more institutionally helpless because they have a restricted involvement within the academic community in the pandemic.” Unclear and needs explaining. 

Response: We have reformulated to Discussion text and presented an explanation for this point.

This paragraph does not make clear what data it draws on: 

“The northern region of Brazil was initially the most affected by the pandemic [67]. In addition, it receives less help and is the region with most economic, political, social adversities, which can influence the performance of students’ resilience levels as they have encountered a period of intense adversity in this region. Similar data have been found in northern Italy [66].” If this is not backed up by the authors’ data, why is it relevant in the discussion? 

Response: We have reformulated the Discussion and emphasized the main results.

There are other similar expressions that I find difficult to understand. (Existem outras expressões semelhantes que acho difícil de entender).

Response: We have revised the translation of the text with the help of a native speaker.

The main problem I see in the discussion is that a large section of it reads like an extension of a literature review. Surprisingly little reflection and reference is given to the authors’ data, considering the significance of the sample size. I would expect much more comparing and contrasting in reference to the authors’ data segments, e.g. gender differences, academic vs low SES demographic comparison, disciplinary comparison, perhaps institutional comparison. I agree with Reviewer 1 that much deeper insights should be gained from such rich data. Unfortunately, the analysis, at least in the way it’s presented in the discussion, remains fairly superficial and doesn’t know clearly add novel insights.

Response: We have reformulated the entire text of the Discussion and presented insights and considerations on the main findings. We have responded to your suggestion and have presented comparisons between gender differences, academic characteristics versus low socioeconomic status, and disciplinary and institutional differences.

The “limitations and strong points” section is weak and seems unnecessary. You could add more value by proposing a number of programs, workshops or offerings universities could add to support PhD students’ resilience building.

Response: We have removed the limitations and strengths section and included suggestions for programs and strategies for universities in the Discussion.

Reviewer 3:

This study provides a large cross-sectional study to assess the impact of the COVID-19 pandemic on postgraduate students in Brazil. The authors should be commended in collecting and analysing a large body of data that provides a snapshot of some key instruments for analysis to assess the impact of fatigue and resilience. I can see from the response to the reviewers that the authors have made a considerable effort to review and adjust based on the feedback provided. Having not read the original submission, I can only comment directly on the revision that has been submitted.

Response: Thank you for your comments and time spent reading this version.

Overall, the manuscript has had much more description and link to previous studies, which does enhance the study, however, in places, the information does lack coherence and flow. There seem to be some paragraphs that are quite short (see first paragraph of introduction) that could be linked better throughout. The 7th paragraph of the introduction, the final sentence needs re-wording – “Literature data” is not correct to refer to and you should include what the significant effects were (greater/improved?). 

Response: We have reformulated the paragraphs of the introduction and revised the entire text, aiming to give it greater coherence and flow. We have readjusted the seventh paragraph of the Introduction.

Your overall aim, in addition to analysing the level of fatigue and resilience, is to also consider the association with sociodemographic and academic factors, which I feel is what is lacking in the discussion. You do a good job of discussing the literature around fatigue and resilience, however, given you present two key figures associated with demographic data, I would like to see more detail around this. The novelty of your paper is that you are assessing Brazilian students – the demographics play a significant role here and this is the novelty in my opinion. This will allow you to utilise the nice studies you have to support your findings in other countries. 

Response: We have rewritten the entire text of the Discussion and have presented insights and thoughts on the main findings. We have included considerations on sociodemographic and academic factors.

Whilst fatigue is not my field, I am curious as to why both a PFS and a VAFS scale are used? I may be misinterpreting this, but they seem to give slightly differing results? You also only seem to primarily discuss the PFS results. I also was unclear (for VAFS) what best result versus worst result meant? If these are common terms for these scales, please ignore, if not, please edit what this means.

Response: We have included a justification in the Methods section “We have chosen these two instruments (PFS and VAFS) to assess fatigue, seeking to ensure a more global and reliable analysis”. The use of more than one instrument to assess the same phenomenon is not uncommon in the literature. The results of the two scales were very close to the PFS, with an average level of 6.08, and the VAFS with an average level of 6.10. What was slightly different was the inferential analysis in which the PFS seemed to be more sensitive to the characteristics that may be associated with fatigue (Source of income). We have modified the description of the result for VAFS.

In 2.3 you refer to n number as (n), however in each table it is referred to as N – please edit. 

Response: The alteration has been made.

Paragraph prior to table 2 starts with “The students” – I assume you mean Most students? Or “The survey found that…”. 

Response: The alteration has been made.

The figure legends for the 2 images are underwritten. Whilst it is good you have included the source, you should also provide the reader with a description of what the images represent. For example, darker colour means more fatigue etc. I also recommend that figure 1 be given an A and B for each image to help the reader.

Response: We have included an explanatory note for each figure and have identified the fatigue figures.

The discussion is generally good (see comments above), but I do feel it lacks flow. I am also slightly confused with paragraph 2, where you refer to females having family commitments may have significant affects on their careers, however in your study the majority of respondents were female and did not have children? Please review this data in relation to your outcomes. It would also be interesting to see what % of the female respondents did have children. 

Response: We have reviewed the data and presented the % of women who had children. We have rewritten the Discussion and improved the analysis of gender differences.

Similar to a previous reviewer, there is mention of many measures currently in place – I suggest you restructure this paragraph to be towards the latter section of the discussion in order to link this in with what you are also suggesting is implemented across the country. For your age data, you mention this differs from the current literature – do you have a suggestion as to why your data may differ? 

Response: We have restructured the Discussion as suggested and left the strategies and measures at the end of the section. We have added an insight into why our age data diverged from a pre-pandemic study “We think that in the pandemic young students had fewer of their own and/or institutional resources to deal with psychic stressors. The lack of a support network and social interactions for many young people may have been risk factors for fatigue”.

4.1 – I’m not sure you require the strong points section – this should be the focus of the discussion and come through to the reader without having to state it.

 Overall, whilst the language has been improved, I would suggest further proof reading and structural/grammar checks prior to resubmission. Thank you for your interesting study and I wish you well with the revision.

Response: We have removed the limitations and strengths section and included it in the Discussion. We have revised the translation with a native speaker.

---

## [Editor Report · Decision Letter 2]

25 Apr 2023

PONE-D-21-28536R2Fatigue and resilience in Master's and PhD students in the Covid-19 pandemic in Brazil: a cross-sectional studyPLOS ONE

Dear Dr. Valóta,

Thank you for submitting your manuscript to PLOS ONE. After careful consideration, we feel that it has merit but does not fully meet PLOS ONE’s publication criteria as it currently stands. Therefore, we invite you to submit a revised version of the manuscript that addresses the points raised during the review process.

 Please see attached file.

We look forward to receiving your revised manuscript.

Kind regards,

Pauline M. Ross, PhD

Academic Editor

PLOS ONE

Additional Editor Comments:

PONE-D-21-28536

"Fatigue and resilience in Master's and PhD students in the Covid-19 pandemic in Brazil: a cross-sectional study"

When this manuscript was submitted – it was sent out for review – because it was so novel and original. The review has now been a long process, with over four reviewers at different times. One reviewer suggested a rejection, two suggested a major review and one a minor review. Each time I have suggested a major review – in light of the reviewer’s comments. Each time the revised version has improved, but not significantly enough to allow a change my views to a minor revision or preferably an acceptance. On each occasion because of the continued novelty and timeliness of the manuscript I have also tried to provide feedback which would be helpful to try and seek an improvement in the manuscript so that it can move towards being published.

I have decided with this final version to do the review myself – in an effort to provide the detail which I hope you may be able to use to bring this manuscript to a conclusion.

As indicated in my last set of comments – this manuscript needed a rigorous review. It still does.

These are my comments from the previous review:

Could you please make the changes as required by the reviewer so that this review can be completed. I notice the reviewer comments that there are sections which still are not logical in flow. This is very important to fix in this version. Following the revision, I will complete what I hope is a final read and edit – so that this manuscript can be resolved. This very much depends however on the quality of the next revised version. I encourage you to persist and perhaps obtain some assistance to produce a quality manuscript. Best wishes and I look forward to the resolution.

This was the response: We have made substantial changes to the manuscript in response to reviewers' requests and presented an improved version.

Unfortunately although the response from the authors was positive, this 3rd review has not gone far enough. There remains several coherency and language issues – which I have detailed below. There are still some interpretation issues – and general formatting of tables and figures which require further attention.

After considerable reflection, I have made the decision to allow one more major revision. This will be the final review. If the authors decide to review again – then please seek English language assistance in the next review of the manuscript. If the manuscript does not improve considerably in language and formatting – then I will need to reject the next version. I really don’t want to reject this manuscript, given the considerable time it has taken to review – and the efforts of the authors. I urge you to make sure the next version – has dealt with the issues described here and in previous reviews. I start with the reviewers comments which I do not believe have been adequately addressed and move onto more specific comments from this version of the manuscript.

Reviewer comments which have not yet adequately addressed:

1. I am also slightly confused with paragraph 2, where you refer to females having family commitments may have significant affects on their careers, however in your study the majority of respondents were female and did not have children? Please review this data in relation to your outcomes. It would also be interesting to see what % of the female respondents did have children.

2. I have re-read the Discussion again and have further points for authors to consider: The writing still needs a major revision throughout the manuscript, perhaps with the help of a Native English speaker?

3. The main problem I see in the discussion is that a large section of it reads like an extension of a literature review. Surprisingly little reflection and reference is given to the authors’ data, considering the significance of the sample size. I would expect much more comparing and contrasting in reference to the authors’ data segments,

Detailed feedback

Introduction

Lines 75-76 Delete from “before and in…. - up until financial” in line 76. This should then read – “marked by psychological pressure with instability” and insert “with”

Line 77 replace “to” with “in”, delete “their teaching and research activities” and replace with “guidance and financial instability”.

Line 79 insert “and” between “experiments interrupt”

Line 83 delete “can lead” and replace with “have been found to”

Line 87 Replace “a” with “for example, a”

Line 88 Make “experiences” experience”

Line 89 Insert “Also” before “In”

Line 92 Delete “Given this scenario” and start at “Fatigue”

Line 113 Delete “data” and replace with “findings”

Line 120 Delete “the” and replace with “being that”

Line 121 Delete “that” and “and” and replace “Authors” with “Studies”

Line 122 Replace “state” with “have found” and delete from “has…… up until adaptive capacity” and replace with “and” and then replace “and” with “are”

Line 123 Replace “helping” with “required” and “seen as a” with “is also”

Line 124 Delete “multidimenionsal construct” and “by” and “among different elements” so that this reads “defined as the interaction between the individual and”

Line 125 delete “contains such as”

Line 126 Insert “also” in between can affect”

Line 138 Delete “probably”

Lines 133-134 Delete “and to verify” and replaced with “and to determine”

2.1 Population and recruitment.

Lines 140 Delete Stricto sensu – as this needs translation – and instead use “only those” and delete from “all areas of knowledge” – as the meaning is unclear. Move up “over 18 years of age from line 141 and put after “graduate students” and insert “who” “and” so that this reads “only those graduate students who were 18 years and over and enrolled in Brazilian educational institutions and responded to all survey items were included in this study”.

Line 142 insert “in the survey and delete “the research”

Line 145 Replace “were” with “was”

Line 147 Delete “subsequently” and “the”

Line 152 Delete “constructed by the authors” and start at “This” and add “collected sociodemographic and academic data”

Line 156 Delete “the phase of greatest impact” as meaning is unclear

Line 161 replace “assess” with “assessed” – past tense

Line 167 Move “as well as total score” to later in the text.

Line 169 Replace “is” with “was – once again should be past tense

Line 184 Delete “addresses resilience and is” and replace with “was” and on line 185 insert “resilience and “ before “levels”

3. Results

Line 224 replace “forms” with “survey”

Line 266 replace “having” with “had” – past tense required. Insert :”were” before “without”

Line 242 replace “from” with “enrolled”

4. Discussion

This need a complete reflective re-write as indicated several times by reviewers. Comments below are for English language – but I have not completed the entire re-write of this discussion – as this is up to the authors – and goes beyond what is reasonable.

Line 357 Replace “assesses” with “assessed

Line 369 Delete “students with difficulties in the two impacted stages showing”

Line 317-377 needs a complete re-write. Start with “delete “in the literature and we found that indicate the” and replace with “Other studies have found a”

Line 372 Replace “investigations” with “studies” to be consistent with previous sentence

Line 373 Insert after “fatigued” “ prior to the COVID-19 pandemic” and delete “this”

Line 374 Delete “this challenging scenario” and replace with “during the COVID-19 pandemic”

Line 374 Move up from line 377 “similar to our results” and place after “hand,”

Line 385 Delete “also” and change ”work” to “worked”

Line 390. Add full stop after [44} and start with “such change created “ and delete “bringing” and delete “the development or an” and replace with “on top of already”

Paragraph from lines 393- 401 requires a complete re-write. The changes are so many thay it is easier for me and perhaps clearer for the authors to understand – if I just write out what should be there.

So replace lines 393-406 with the following text:

In public universities in Brazil there is a constant demand for academics or researchers to increase their scientific production as they are under pressure from government classification processes which rank institutions and graduate programs on their research productivity. Researchers or academics must then prioritise their work into teaching, research and extension activities. If they are to be promoted and be competitive for jobs then they must produce publications and have success in research funding. This pressure and bureaucratic structure can cause an overload on academics which is transferred to and has a direct impact on graduate students

This is in contrast to private universities in Brazil where there is less demand for research publications and a greater focus on teaching. Academics are often on more balanced workload allocations with contracts which are not solely based on research. Graduates students also are in better financial situations and this is correlated with resilience.

It is really beyond my role to rewrite the rest of the text from lines 405-466. This needs however a complete re-write following what I have written above. I have not re-written pages 23-25 lines 426-486 – need careful re-write to bring out the main points.

However – a brief look

Line 425 – delete the entire line as this does not make sense.

Line 437 – more issues with past tense – this should be “were” delete “are”

5. Conclusion

Lines 481-486 Re write from lines 481-487.

Final comments to authors - please review this manuscript in detail - and ensure your efforts to date can be used to bring this to a positive conclusion.

---

## [Author Response · Author response to Decision Letter 2]

1 Sep 2023

Academic Editor

When this manuscript was submitted – it was sent out for review – because it was so novel and original. The review has now been a long process, with over four reviewers at different times. One reviewer suggested a rejection, two suggested a major review and one a minor review. Each time I have suggested a major review – in light of the reviewer’s comments. Each time the revised version has improved, but not significantly enough to allow a change to a minor revision or preferably an acceptance. On each occasion because of the continued novelty and timeliness of the manuscript I have also tried to provide feedback which would be helpful to try and seek an improvement in the manuscript so that it can move towards being published. 

I have decided with this final version to do the review myself – in an effort to provide the detail which I hope you may be able to use to bring this manuscript to a conclusion.

As indicated in my last set of comments – this manuscript needed a rigorous review.

Response: From the outset, we have tried to meet all the requests from all the reviewers. I believe that some questions have been difficult due to a lack of understanding of the request, which may be due to the difference in language and also the way in which it is written. In this version we have made every effort to meet the editor with very pertinent suggestions.

Unfortunately although the response from the authors was positive, this 3rd review has not gone far enough. There remains several coherency and language issues – which I have detailed below. There are still some interpretation issues – and general formatting of tables and figures which require further attention.

After considerable reflection, I have made the decision to allow one more major revision. If the authors decide to review again – then please seek English language assistance in the next review of the manuscript. If the manuscript does not improve considerably in language and formatting – then I will need to reject the next version. I really don’t want to reject this manuscript, given the considerable time it has taken to review – and the efforts of the authors. I urge you to make sure the next version – has dealt with the issues described here and in previous reviews. I start with the reviewers comments which I do not believe have been adequately addressed and move onto more specific comments from this version of the manuscript.

 Response: Thank you for this opportunity to review the manuscript and for your comments on any changes.

Reviewer comments which have not yet adequately addressed:

1. I am also slightly confused with paragraph 2, where you refer to females having family commitments may have significant affects on their careers, however in your study the majority of respondents were female and did not have children? Please review this data in relation to your outcomes. It would also be interesting to see what % of the female respondents did have children. 

 Response: We have reviewed the data and presented the % of women who had children. The change is in the lines: 216-217. We have changed the discussion and emphasized the finding based on our data. The change is in the lines: 407-416. 

I have re-read the Discussion again and have further points for authors to consider: The writing still needs a major revision throughout the manuscript, perhaps with the help of a Native English speaker? 

Response: We have revised the translation of the text with the help of a native speaker.

The main problem I see in the discussion is that a large section of it reads like an extension of a literature review. Surprisingly little reflection and reference is given to the authors’ data, considering the significance of the sample size. I would expect much more comparing and contrasting in reference to the authors’ data segments. 

Response: We have amended the discussion based on the suggestions and the suggested model paragraph. /Alteramos a discussão com base nas sugestões e no modelo de parágrafo que foi sugerido.

Detailed feedback: 

Introduction 

Lines 75-76 Delete from “before and in…. - up until financial” in line 76. This should then read – “marked by psychological pressure with instability” and insert “with”. 

Response: The alteration has been made. 

Line 77 replace “to” with “in”, delete “their teaching and research activities” and replace with “guidance and financial instability”.

Response: The alteration has been made. 

Line 79 insert “and” between “experiments interrupt”. 

Response: The alteration has been made. 

Line 83 delete “can lead” and replace with “have been found to”

 Response: The alteration has been made.

Line 87 Replace “a” with “for example, a” 

Response: The alteration has been made.

Line 88 Make “experiences” experience” 

Response: The alteration has been made.

Line 89 Insert “Also” before “In”

Response: The alteration has been made.

Line 92 Delete “Given this scenario” and start at “Fatigue”

Response: The alteration has been made.

Line 113 Delete “data” and replace with “findings” 

Response: The alteration has been made.

Line 120 Delete “the” and replace with “being that” 

Response: The alteration has been made.

Line 121 Delete “that” and “and” and replace “Authors” with “Studies” 

Response: The alteration has been made.

Line 122 Replace “state” with “have found” and delete from “has…… up until adaptive capacity” and replace with “and” and then replace “and” with “are”

Response: The alteration has been made.

Line 123 Replace “helping” with “required” and “seen as a” with “is also” 

Response: The alteration has been made.

Line 124 Delete “multidimenionsal construct” and “by” and “among different elements” so that this reads “defined as the interaction between the individual and” 

Response: The alteration has been made.

Line 125 delete “contains such as” 

Response: The alteration has been made.

Line 126 Insert “also” in between can affect” 

Response: The alteration has been made.

Line 138 Delete “probably” 

Response: The alteration has been made.

Lines 133-134 Delete “and to verify” and replaced with “and to determine”

Response: The alteration has been made.

2.1 Population and recruitment.

Lines 140 Delete Stricto sensu – as this needs translation – and instead use “only those” and delete from “all areas of knowledge” – as the meaning is unclear. Move up “over 18 years of age from line 141 and put after “graduate students” and insert “who” “and” so that this reads “only those graduate students who were 18 years and over and enrolled in Brazilian educational institutions and responded to all survey items were included in this study”. 

Response: The alteration has been made.

Line 142 insert “in the survey and delete “the research”

Response: The alteration has been made.

Line 145 Replace “were” with “was”

Response: The alteration has been made.

Line 147 Delete “subsequently” and “the” 

Response: The alteration has been made.

Line 152 Delete “constructed by the authors” and start at “This” and add “collected sociodemographic and academic data” 

Response: The alteration has been made.

Line 156 Delete “the phase of greatest impact” as meaning is nuclear 

Response: The alteration has been made.

Line 161 replace “assess” with “assessed” – past tense 

Response: The alteration has been made.

Line 167 Move “as well as total score” to later in the text.

Response: The alteration has been made.

Line 169 Replace “is” with “was – once again should be past tense 

Response: The alteration has been made.

Line 184 Delete “addresses resilience and is” and replace with “was” and on line 185 insert “resilience and “ before “levels”

Response: The alteration has been made.

3. Results 

Line 224 replace “forms” with “survey”

Response: The alteration has been made.

Line 266 replace “having” with “had” – past tense required. Insert :”were” before “without” 

Response: The alteration has been made.

Line 242 replace “from” with “enrolled” 

Response: The alteration has been made.

4. Discussion

This need a complete reflective re-write as indicated several times by reviewers. 

Response: We changed it and tried to write more reflectively.

Comments below are for English language – but I have not completed the entire re-write of this discussion – as this is up to the authors – and goes beyond what is reasonable. 

Line 357 Replace “assesses” with “assessed 

Response: The alteration has been made.

Line 369 Delete “students with difficulties in the two impacted stages showing” 

Response: The alteration has been made.

Line 317-377 needs a complete re-write. Start with “delete “in the literature and we found that indicate the” and replace with “Other studies have found a” 

Response: The alteration has been made.

Line 372 Replace “investigations” with “studies” to be consistent with previous sentence

Response: The alteration has been made.

Line 373 Insert after “fatigued” “ prior to the COVID-19 pandemic” and delete “this” Response: The alteration has been made.

Line 374 Delete “this challenging scenario” and replace with “during the COVID-19 pandemic” 

Response: The alteration has been made.

Line 374 Move up from line 377 “similar to our results” and place after “hand,” 

Response: The alteration has been made.

Line 385 Delete “also” and change ”work” to “worked” 

Response: The alteration has been made.

Line 390. Add full stop after [44} and start with “such change created “ and delete “bringing” and delete “the development or an” and replace with “on top of already” Response: The alteration has been made.

Paragraph from lines 393- 401 requires a complete re-write. The changes are so many thay it is easier for me and perhaps clearer for the authors to understand – if I just write out what should be there.

So replace lines 393-406 with the following text:

In public universities in Brazil there is a constant demand for academics or researchers to increase their scientific production as they are under pressure from government classification processes which rank institutions and graduate programs on their research productivity. Researchers or academics must then prioritise their work into teaching, research and extension activities. If they are to be promoted and be competitive for jobs then they must produce publications and have success in research funding. This pressure and bureaucratic structure can cause an overload on academics which is transferred to and has a direct impact on graduate students

This is in contrast to private universities in Brazil where there is less demand for research publications and a greater focus on teaching. Academics are often on more balanced workload allocations with contracts which are not solely based on research. Graduates students also are in better financial situations and this is correlated with resilience. 

Response: We would like to thank you for the paragraph template, it was very useful for reformulating the whole discussion.

It is really beyond my role to rewrite the rest of the text from lines 405-466. This needs however a complete re-write following what I have written above. I have not re-written pages 23-25 lines 426-486 – need careful re-write to bring out the main points. However – a brief look 

Response: We rewrote the paragraphs carefully and thoughtfully.

Line 425 – delete the entire line as this does not make sense.

Response: The alteration has been made.

Line 437 – more issues with past tense – this should be “were” delete “are”

Response: The alteration has been made.

5. Conclusion

Lines 481-486 Re write from lines 481-487. 

Response: We have rewritten the second paragraph of the conclusion to make it more objective.

Final comments to authors - please review this manuscript in detail - and ensure your efforts to date can be used to bring this to a positive conclusion.

Response: Thank you for the opportunity to review the manuscript once again.

---

## [Editor Report · Decision Letter 3]

2 Oct 2023

PONE-D-21-28536R3Fatigue and resilience in Master's and PhD students in the Covid-19 pandemic in Brazil: a cross-sectional studyPLOS ONE

Dear Dr. Valóta,

Thank you for submitting your manuscript to PLOS ONE. After careful consideration, we feel that it has merit but does not fully meet PLOS ONE’s publication criteria as it currently stands. Therefore, we invite you to submit a revised version of the manuscript that addresses the points raised during the review process.

In general this manuscript has improved and it is very close to acceptance for publication. I would still like to have the authors seek further advice on sentence structure from a native English speaker.  I understand this may have already been done – but it could be repeated.  The discussion reads better than the introduction – but the purpose of the introduction is to engage the reader – and more could be done here.

Specific areas to fix.

Write the abstract as a coherent 300 word statement without subheadings.   Also in the conclusion replace intensify with design – lines 61

See examples on line and the instructions to authors below

The Abstract should:

Describe the main objective(s) of the studyExplain how the study was done, including any model organisms used, without methodological detailSummarize the most important results and their significanceNot exceed 300 words
Add the aims and research question at the end of the introduction – lines 134 – so that the reader understands the purpose of the study clearly.

Line 175 What is meant here by “horizontal line” – do you mean the scale goes from 0-10?  I am not sure how measurements in centimetre are involved?  The one I researched on line is a scale from 0-10 – remove reference to cm.

Line 312 What is income five minimum wages above?  Perhaps you mean “Having an income which is at least five times greater than the minimum wage is correlated with greater resilience ”  Please use this sentence and replace if correct.

Line 314  If these were Master’s students – then why are they retired – what proportion of masters students were retired – is this an important finding?  Perhaps reconsider.

Discussion –

Should mainly be written in the past tense – see line 404 – replace “is” with “was” and check throughout.

Line 424 Commence with “This study found” and delete “It was identified”

Line 437 Replace “are” with “were” – past tense again.

We look forward to receiving your revised manuscript.

Kind regards,

Pauline M. Ross, PhD

Academic Editor

PLOS ONE

Journal Requirements:

Additional Editor Comments:

In general this manuscript has improved and it is very close to acceptance for publication. I would still like to have the authors seek further advice on sentence structure from a native English speaker. I understand this may have already been done – but it could be repeated. The discussion reads better than the introduction – but the purpose of the introduction is to engage the reader – and more could be done here.

Specific areas to fix.

1. Write the abstract as a coherent 300 word statement without subheadings. Also in the conclusion replace intensify with design – lines 61

See examples on line and the instructions to authors below

The Abstract should:

• Describe the main objective(s) of the study

• Explain how the study was done, including any model organisms used, without methodological detail

• Summarize the most important results and their significance

• Not exceed 300 words

2. Add the aims and research question at the end of the introduction – lines 134 – so that the reader understands the purpose of the study clearly.

Line 175 What is meant here by “horizontal line” – do you mean the scale goes from 0-10? I am not sure how measurements in centimetre are involved? The one I researched on line is a scale from 0-10 – remove reference to cm.

Line 312 What is income five minimum wages above? Perhaps you mean “Having an income which is at least five times greater than the minimum wage is correlated with greater resilience ” Please use this sentence and replace if correct.

Line 314 If these were Master’s students – then why are they retired – what proportion of masters students were retired – is this an important finding? Perhaps reconsider.

Discussion –

Should mainly be written in the past tense – see line 404 – replace “is” with “was” and check throughout.

Line 424 Commence with “This study found” and delete “It was identified”

Line 437 Replace “are” with “were” – past tense again.

---

## [Author Response · Author response to Decision Letter 3]

16 Nov 2023

Academic Editor

Thank you for submitting your manuscript to PLOS ONE. After careful consideration, we feel that it has merit but does not fully meet PLOS ONE’s publication criteria as it currently stands. Therefore, we invite you to submit a revised version of the manuscript that addresses the points raised during the review process.

In general this manuscript has improved and it is very close to acceptance for publication. I would still like to have the authors seek further advice on sentence structure from a native English speaker. I understand this may have already been done – but it could be repeated. The discussion reads better than the introduction – but the purpose of the introduction is to engage the reader – and more could be done here.

Response: We have changed the logic and tried to involve the reader in the new version of the introduction.

Specific areas to fix.

 Write the abstract as a coherent 300 word statement without subheadings. Also in the conclusion replace intensify with design – lines 61

Response: The alteration has been made. 

Add the aims and research question at the end of the introduction – lines 134 – so that the reader understands the purpose of the study clearly.

Response: We added the questions associated with the objectives that were already described at the end of the introduction.

Line 175 What is meant here by “horizontal line” – do you mean the scale goes from 0-10? I am not sure how measurements in centimetre are involved? The one I researched on line is a scale from 0-10 – remove reference to cm.

Response: The alteration has been made. 

Line 312 What is income five minimum wages above? Perhaps you mean “Having an income which is at least five times greater than the minimum wage is correlated with greater resilience” Please use this sentence and replace if correct.

Response: The alteration has been made. 

Line 314 If these were Master’s students – then why are they retired – what proportion of masters students were retired – is this an important finding? Perhaps reconsider. 

Response: We have modified the text. This paragraph shows that students who have a source of income from retirement are more resilient and this is important data, as it was a statistically significant variable. Of the 10 students whose source of income comes from retirement, 5 are master's students (50%) and have an average age of 60. 

Discussion –

Should mainly be written in the past tense – see line 404 – replace “is” with “was” and check throughout.

Response: The alteration has been made. 

Line 424 Commence with “This study found” and delete “It was identified”

Response: The alteration has been made. 

Line 437 Replace “are” with “were” – past tense again.

Response: The alteration has been made.

---

## [Editor Report · Decision Letter 4]

20 Nov 2023

Fatigue and resilience in Master's and PhD students in the Covid-19 pandemic in Brazil: a cross-sectional study

PONE-D-21-28536R4

Dear Dr. Valóta,

We’re pleased to inform you that your manuscript has been judged scientifically suitable for publication and will be formally accepted for publication once it meets all outstanding technical requirements.

Kind regards,

Pauline M. Ross, PhD

Academic Editor

PLOS ONE
---

## [Editor Report · Acceptance letter]

23 Nov 2023

PONE-D-21-28536R4 

Fatigue and resilience in Master's and PhD students in the Covid-19 pandemic in Brazil: a cross-sectional study 

Dear Dr. Valóta:

I'm pleased to inform you that your manuscript has been deemed suitable for publication in PLOS ONE. Congratulations! Your manuscript is now with our production department. 

Kind regards, 

on behalf of

Professor Pauline M. Ross 

Academic Editor

PLOS ONE